

# Real-time infectious disease endurance indicator system for scientific decisions using machine learning and rapid data processing

Shivendra Dubey, Dinesh Kumar Verma and Mahesh Kumar

Computer Science and Engineering, Jaypee University of Engineering and Technology, Guna, Madhya Pradesh, India

## ABSTRACT

The SARS-CoV-2 virus, which induces an acute respiratory illness commonly referred to as COVID-19, had been designated as a pandemic by the World Health Organization due to its highly infectious nature and the associated public health risks it poses globally. Identifying the critical factors for predicting mortality is essential for improving patient therapy. Unlike other data types, such as computed tomography scans, x-radiation, and ultrasounds, basic blood test results are widely accessible and can aid in predicting mortality. The present research advocates the utilization of machine learning (ML) methodologies for predicting the likelihood of infectious disease like COVID-19 mortality by leveraging blood test data. Age, LDH (lactate dehydrogenase), lymphocytes, neutrophils, and hs-CRP (high-sensitivity C-reactive protein) are five extremely potent characteristics that, when combined, can accurately predict mortality in 96% of cases. By combining XGBoost feature importance with neural network classification, the optimal approach can predict mortality with exceptional accuracy from infectious disease, along with achieving a precision rate of 90% up to 16 days before the event. The studies suggested model's excellent predictive performance and practicality were confirmed through testing with three instances that depended on the days to the outcome. By carefully analyzing and identifying patterns in these significant biomarkers insightful information has been obtained for simple application. This study offers potential remedies that could accelerate decision-making for targeted medical treatments within healthcare systems, utilizing a timely, accurate, and reliable method.

## INTRODUCTION

The unusual COVID-19 outbreak, caused by SARS-CoV-2 (severe acute respiratory syndrome coronavirus 2), began in Wuhan, China, in December 2019. As of April 24, 2021, the COVID-19 pandemic caused by the highly contagious SARS-CoV-2 virus had been rapidly spread worldwide, resulting in a cumulative count of over 144 million positively verified cases, and the number of confirmed fatalities resulting from the examined condition or disease exceeds 3 million, as reported by the WHO. Amidst that

Corresponding author
Shivendra Dubey,
shivendrashivay@gmail.com

pandemic, estimating mortality ratios relies on computationally derived and surveillance-based approaches, often using basic methodologies. Consequently, a significant variation exists in the estimated case mortality rates across different countries. Determining the comparative risk of mortality among patients and how to allocate medical resources during the pandemic is aided by identifying the major biomarkers of mortality. The ordinary flu's symptoms, such as coughing, runny noses, and fever congestion, are strikingly similar to those of an infectious disease like COVID-19 (*Ahmed & Jeon, 2022*). As the pandemic expanded, additional signs and symptoms arose, including loss of taste and smell (*Sevillano et al., 2021*; *Makaremi et al., 2022*). Severe cases could cause pneumonia and other respiratory illnesses (*Mousavizadeh & Ghasemi, 2021*; *Li et al., 2022*; *Ray et al., 2021*). It is difficult to determine which disease-affected characteristics have a bigger effect on patient death because there is a vast range of them. With machine learning, enormous data sets can be quickly and reliably analyzed for patterns that can be used to create models that identify risk factors (*Mousavizadeh & Ghasemi, 2021*).

From unaffected to severely infectious diseases like COVID-19 in individuals to respiratory illness, COVID-19 exhibits a wide range of health-related indications (*Park et al., 2022*; *Ahmad, Ahmed & Jeon, 2022*). During that phase of the global outbreak, an unanticipated rise in infectious disease cases placed tremendous strain on systems of healthcare, resulting in a scarcity of assets for critical care. The majority of patients who adapted to healthcare facilities had recovered, but certain individuals suffer severe respiratory difficulties that necessitate the use of respirators. Furthermore, a large number of these ventilator-dependent people pass away due to quickly worsening respiratory disorders. When prioritizing patients for hospitalization, knowing the probability of dying will prove essential. This could be achieved by discovering critical biomarkers in the blood of fatalities from infectious diseases like COVID-19. In addition, determining the primary indicators of morbidity offers direction over assigning scarce assets to patients in need of instantaneous ventilation and who are in a critical position of passing away. Within this framework, machine learning (ML) can help by analyzing large datasets using multiple features to rapidly recognize trends and create ML algorithms that can precisely evaluate danger indicators related to the degree of infection. ML methods have also been instrumental in the identification of infectious disease cases based on CT scans and chest X-ray images, but blood biomarkers are more effective at early predicting the mortalities due to infectious disease.

For identifying as well as diagnosing infections among patients, several researchers have used machine learning algorithms (*Makaremi et al., 2022*). With their assistance, RT-PCR (reverse transcription polymerase chain reaction) at the societal level can be performed more quickly during a pandemic. Most of the ML-driven investigations of infectious diseases like COVID-19 are built upon CT scans or chest x-rays (*Park et al., 2022*). In an environment with few resources, it is challenging to gather this data (*Ahmad, Ahmed & Jeon, 2022*). Therefore, a model based on data gathered utilizing affordable and accessible tests is required. Cough samples have shown for the identification of the infectious disease like COVID-19 (*Peker et al., 2022*). To accurately determine the severity of SARS-CoV-2 infections, scalable technologies are necessary. Researchers utilized multiple ML models to

forecast the probability of experiencing severe complications and fatality due to that infectious disease (*Li, Chen & Zeng, 2022*). Two distinct models were used to predict infected patient deaths based on their clinical and scientific characteristics (*Ahmed et al., 2022*).

A real-time "endurance indicator system using machine learning and rapid data processing" is chosen due to its numerous advantages over traditional methods of analyzing pandemic-related data. Firstly, the use of machine learning algorithms enables more precise real-time predictions of the pandemic's endurance indicators. This can be helps to decision-makers to make informed decisions about implementing public health measures and adapting existing policies to respond to the rapidly evolving nature of the pandemic. Secondly, the rapid data processing techniques employed in the system enable the generation of real-time indicators of the pandemic's endurance. This enables decision-makers to access timely up-to-date information on the state of the pandemic and respond quickly to emerging trends or outbreaks. Lastly, the integration of multiple data sources and advanced data analysis techniques leads to a more comprehensive and accurate understanding of the pandemic's endurance. This empowers decision-makers to make informed and timely decisions based on a complete picture of the pandemic's impact, rather than relying on limited or outdated data. Overall, the use of this approach provides a powerful and effective tool for decision-makers dealing with the multifaceted challenges posed by the infectious diseases pandemic.

## LITERATURE REVIEW

In the COVID-19 scoring system, patients can be classified into high- and low-risk groups. Those in the high-risk group are subjected to significantly higher mortality risks than their low-risk counterparts (*Myoung, 2022*). In this logistic regression coefficients, feature analysis, multivariable analysis, lasso binary, and a prediction model were produced. A model that exhibits robust discriminative control and the independently validated model boast a 94% AUC, signifying its reliable predictive capability and robustness. It was developed by leveraging eight investigated clinical factors, including blood and age parameters. There are some parameters like $SpO_2$, age, lymphocyte count, and LDH (Lactate De-Hydrogenase) are shown to be a group of crucial parameters to produce a mortality prediction model (*Bagabir et al., 2022*). The classification problem was tackled using multivariable logistic regression, and the additional validation dataset resulted in an AUC of 97%. They used this to develop a nomogram that could predict death. Models used for predicting mortality using the Denmark and UK's COVID-19 dataset; that had AUCs of 0.907 during diagnosis, 0.721 following ICU admission, and 0.819 at access to the hospital. There some factors identified like age, hypertension, and body mass index that are typical risk factors (*Arslan & Arslan, 2021*).

The XGBoost-based model was created after analyzing blood samples from around 485 victims in Hubei, China. The lymphocytes, hs-CRP, and LDH characteristics were three of the most important in the suggested medically operational single tree XGBoost model. These three qualities, as well as their threshold, have been used to develop the decision-making processes iteratively. Including a minimum of 90% daily accuracy provided a clear

machine learning approach (*Meraihi et al., 2022*). These models' consistency was found to be 94% accurate 7 days before the outcome (*Jia, Chen & Lyu, 2022*). But their findings could be skewed. The approach employed to assess consistency was biased in favour of the large number of samples taken close to the outcome day.

The F1-score exhibits stability as it fluctuates the results from day 1 to day 17, which is 0.968 to 0.689. The results are also unreliable because of the uneven test set. Accurate early-stage predictions are preferable for developing early treatment plans. A machine learning process suggested getting around these fatality prediction problems and boosting performance. We looked at the provided dataset, which contained the biomarkers discovered through blood tests, to create our models (*Khan, Khan & Nazir, 2022*). We devised a solution by leveraging neural networks (NNs) for classification and prioritizing XGBoost feature importance. Overfitting can occasionally result from various characteristics in a small cohort. There are the most important biomarkers for predictions were chosen as a potent combination of five traits. Additional information about the features is also provided by the feature's graph analysis, which demonstrates trends in evolution. Numerous methods and thorough testing were used to build a solid foundation of confidence in the model. Our model is remarkably precise and reliable during a patient's illness. With fewer traits and greater certainty, this would facilitate a quicker diagnosis. Healthcare systems can use the approach to focus resource usage and improve treatment plans (*Sartorao Filho et al., 2022*; *Paul et al., 2023*).

For COVID-19, ML created some models to forecast respiratory collapse 48 h after patient admission. The XGBoost model achieved the greatest mean accuracy and AUC, which were 0.920. The emergency department's type of oxygen supply, the demographic, emergency severity index score, serum lactate, respiratory rate, and features were the factors that had the most impact. Most of the studies haven't shown how consistently their findings hold up over time. This analysis is essential to determining how well the model performs and how well it can be used in practical situations when a patient is ill (*Harikrishnan, Pranay & Nagaraj, 2022*; *Rohaim et al., 2020*).

Machine learning algorithms have been used to predict COVID-19 along with mental health disorders, such as depression, anxiety, and stress, using online survey data collected from both the general population and rajyoga meditators. The selected algorithms, including naive bayes, decision trees, random forests, K-nearest neighbors, and support vector machines, were chosen due to their proven accuracy in predicting psychological disorders. These findings can help improve early intervention and diagnosis for sufferers (*Irene & Beulah, 2022*). The COVID-19 pandemic has led to considerable disruptions across multiple financial domains, prompting substantial alterations in the prevailing economic climate and social activities across the globe. However, it has also led to a surge in research aimed at mitigating the adverse effects of the virus. This research has been conducted both outside and within clinical settings to assist communities in overcoming the challenges posed by the pandemic (*Chamseddine et al., 2022*). With the rapid evolution of the virus and its effects, accurate and affordable diagnostic tools are crucial in delivering

effective healthcare services to the masses. Adapting biomarkers and digital medicine technology can provide better prospects for public health management. However, multidisciplinary research, specific policy, and technology adaptation are essential for developing cost effective diagnostic tools with high specificity. By adopting these measures, we have overcome the difficulty in the vaccine, drug development, and diagnosis process to limit COVID-19 spreads (*Jalaber et al., 2020*). Table 1 represents the various models which are work on the fatality prediction of infectious diseases like COVID-19 on the based on blood biomarkers.

The COVID-19 pandemic has rendered digital technologies indispensable for both economic and social health. In response, there has been a surge of effort devoted to developing modules and creating applications that can extract valuable insights from vast databases. An epidemic has the potential to take many forms and can be quite beneficial. Whether it is contact tracing, remote work, telemedicine, or other applications, digital technologies are essential tools in the fight against infectious disease like COVID-19. By harnessing the power of these technologies, we can better manage the pandemic and work towards a healthier, safer future (*Hernandez-Matamoros et al., 2020*). A developed framework is utilized to interpret the prediction of outcomes to achieve explainable AI. The proposed prediction system has been deployed on a website, which allows users to get an immediate prognosis of illness based on their symptoms. This system can be a valuable tool for healthcare providers to make accurate and quick decisions in identifying infected patients and providing appropriate care (*Lee & Kim, 2022*).

Simulated neuronal programs have been leveraged in conjunction with artificial intelligence to forecast the likelihood of survival in COVID-19 sufferer exhibiting heart dysfunction. AI algorithms, including deep learning, cognitive computing, and machine learning, have been used to develop a treatment protocol for a successful evaluation. With cardiovascular disease being a major cause of death and expensive to treat, AI can assist in recognizing new drugs and improving and treating the efficacy of clinicians. AI is becoming an increasingly accepted feature of various healthcares and engineering sectors, providing a promising platform for viable treatment (*Zhan et al., 2021*).

The application of AI has positively impacted the healthcare system during the pandemic by improving disease dynamics and diagnosis. The compilation of novel coronavirus-targeted drugs and gene biomarkers identified through ML approaches provides sufficient evidence to support the use of AI in the COVID-19 study. While technology advances, the use of artificial intelligence in healthcare is expected to continue to improve and grow patient outcomes (*Rădulescu, Williams & Cavanagh, 2020*). An innovative methodology founded on deep learning techniques, designed a hybrid single-window system that amalgamates LSTM and CNN models, to forecast and analyze COVID-19 in real-time using Twitter data. The application of this approach has the potential to yield new knowledge and resources for monitoring and controlling the propagation of the disease (*Malavika et al., 2021*). A taxonomy and discussion present the importance of machine learning for enhancing telemedicine during the COVID-19 pandemic (*Prem et al., 2020*; *Grzybowska et al., 2022*; *Dubeya, Kumar & Verma, 2022*).

**Table 1 Studies on infectious disease endurance indicator system based on blood biomarker.**

| S. No. | Algorithms | Biomarker | F1-Score | Accuracy | AUC (Area under curve) | References |
|---|---|---|---|---|---|---|
| 1 | Multivariate and univariate binomial based approach | Erythrocytes, ferritin, Eosinophils, and | NA | 72.31–79.41% | 91.5% | *Li et al. (2022)* |
| 2 | ANN (Artificial Neural Network) and RF (Random Forest) based approaches | Full blood counts Aspartate Aminotransferase, hs-CRP (High Sensitivity C-Reactive Protein), the platelets and WBC (White Blood Cells) counts | NA | 90.00–91.01% | 94.01–95.11% | *Myoung (2022)* |
| 3 | TWRF (Three Way Random Forest) and RF (Random Forest) approach | LDH (Lactate Dehydrogenase), AST (Aspartate Aminotransferase), ALP (Alkaline Phosphatase), GGT (Gamma-Glutamyl Transferase), ALT (Alanine Transaminase) plasama level | NA | 82.10–86.00% | 84.00–86.00% | *Javed et al. (2023)* |
| 4 | Nomogram with the use of Multi-tree XGBoost and self Multi-tree XGBoost | Age, LDH, hs-CRP, lymphocyte (%),neutrophils (%) | NA | 100% | 99.1% | *Dubey, Verma & Kumar (2024)* |
| 5 | GBDT (Gradient boosting decision tree) | 27 routine lab results with race, sex, age (Patient demographic features) | NA | NA | 85.4% | *Grzybowska et al. (2022)* |
| 6 | MLR (Multiple Logistic Regression) | NT-proBNP (N-terminal pro-brain natriuretic peptide) and GFR (Glomerular filtration rate) | NA | NA | NA | *Dubeya, Kumar & Verma (2022)* |

## MATERIALS AND METHODS

The proposed methodology aims to develop a robust system, which can provide decision-makers with real-time indicators of the endurance of the COVID-19 pandemic. To achieve this goal, the methodology integrates advanced data analysis techniques and machine learning algorithms.

### Preprocessing and dataset

The only source of data for feature analysis, testing, and training, this dataset consists of 2,779 computerized patient records from Tongji Hospital in Wuhan, China, that are the cases confirmed or probable COVID-19. The original dataset included data on admission time, sample time, outcome (death or survivance), and discharge time, comprising 375 COVID-19 cases, including 74 biomarkers. This data may include:

**Patient demographics:** name, age, gender, address, contact information, *etc.*

**Medical history:** previous diagnoses, surgeries, allergies, medications, family history of diseases, *etc.*

**Clinical information:** lab test results, imaging studies (such as X-rays, CT scans, and MRIs), vital signs (such as blood pressure, heart rate, and temperature), diagnosis, treatment plan, and outcomes.

**Hospitalization information:** admission and discharge dates, length of stay, reason for admission, *etc.*

**Social and behavioural information:** lifestyle habits, smoking, drinking, *etc*.

**Research-related data:** patient participation in clinical trials, research studies, *etc*.

It is essential to note that patient data is highly sensitive and should be handled with care to ensure confidentiality and privacy. The data can be used to analyze patterns of disease prevalence, risk factors, and treatment outcomes, which can help improve patient care and inform public health policies.

For testing and training, they only used information from each patient's final samples (*Meraihi et al., 2022*). In our investigation, an instance from every day of every patient has been used for testing and training purposes. There were several rows for each patient, each containing readings from a different date.

Furthermore, on certain days multiple tasks outcome has taken place at varied intervals. Each row showing the same patient's observations which were combined to generate a unique data point corresponding to each patient's specific day of interest. When features have several recordings made at different times within a single day, the reading received first on that day is considered when constructing the single consolidated statistic. The model quickly learned the most useful features in predicting mortality; the training set's missing values were imputed through application of the k-nearest neighbor (KNN) method. The imputed value was obtained through a weighting scheme that prioritized the 10 nearest neighbors with weights determined by the Euclidean distance method. Using the min-max approach, the training data was scaled. This scaling method was used to preserve statistical data point structure points within every feature because most characteristics do not match a normal distribution.

## Splitting data for individual patient segregation

We randomly selected 80% of the total population and formed the training set. We reserved the remaining 20% for the testing set to ensure patient exclusivity. Since incorporating data points from one patient in both the test and training sets could introduce bias, Figs. 1 and 2 depict the distribution of classes as trained data and tested data, respectively, according to the days leading up to the outcome. The pre-processed dataset for analysis comprises 1,766 data points, corresponding to 370 unique patients. Of the patient cohort, 54.3% survived COVID-19, while 45.7% succumbed or were dead. Each patient's disease progression was tracked over time, resulting in a longitudinal dataset encompassing between one and twelve data points per patient. These data points collected several days before either of the two possible outcomes-death or survival (Fig. 2). Consequently, the dataset provides a comprehensive view of the temporal evolution of SARS-CoV-2 in patients, offering valuable insights for analysis and modeling. We treat each patient's reading as a separate data point for further analysis because we want to develop a fatality forecast distinct from the number of days until the outcome. The outcomes of this study are contrasted with those that do not use KNN imputation. A data point in a high-dimensional space can be matched, including its closest k-neighbours, using the KNN method.

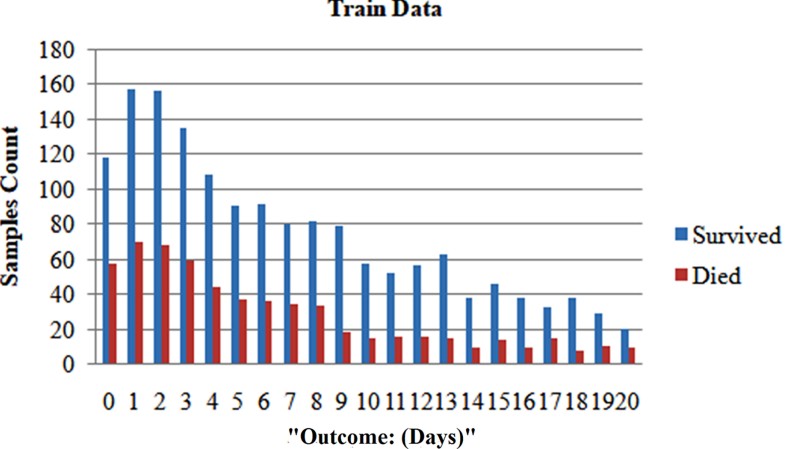

**Figure 1** After dividing, the two classes' distribution in the train set.

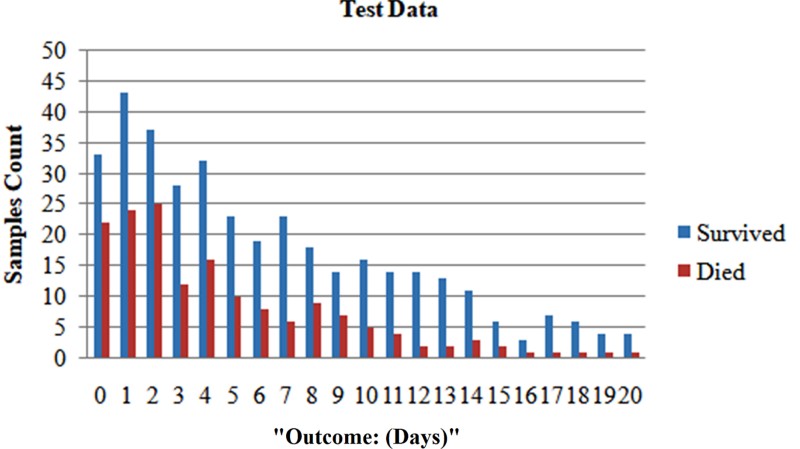

**Figure 2** After dividing, the two classes' distribution in the test set.

**Imputed test set:** The important information about the chosen five characteristics from their 10 closest neighbours within the train set was used to impute each data point inside the testing set. In the end, 71 patients provided 213 samples for the test set. A solid representative of both classes can be seen in the test set, as seen by the total number of dead patients assigned to 0.563.

**Non-imputed test set:** These algorithms' efficacy under natural conditions, without the influence of artificially generated data was assessed. We removed rows of missing data for any five characteristics and allotted them to the test set. The test set consisted of 115 samples obtained from 65 patients, resulting in a patient-care ratio of 0.513.

## Machine learning pipeline

The whole pipeline employed in this study to carry out the objective of predicting mortality is shown in Fig. 3. All models for predicting mortality have already been trained to utilize

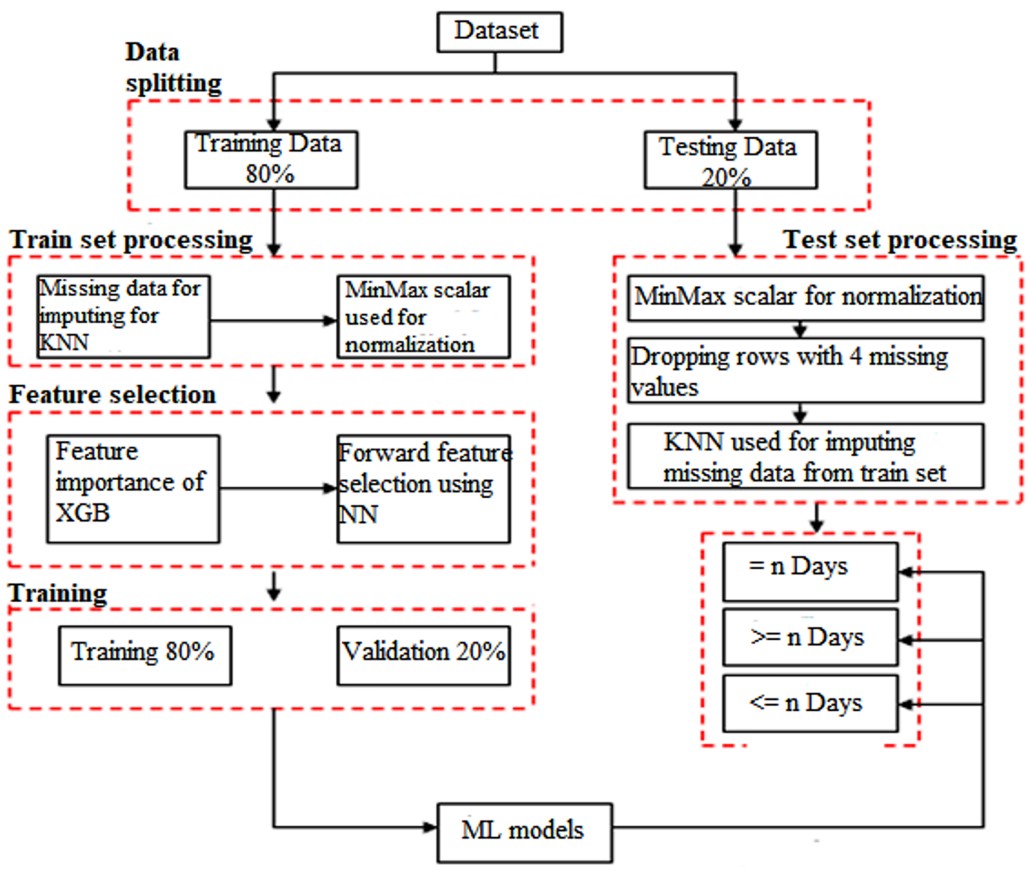

**Figure 3** Diagram showing the pathway for developing the models utilized in this investigation.

samples from all days in this database, no matter how many days until the outcome. Following the processing of the data, the XG-Boost classifier was used to estimate the relevance of the features, and a neural network was used to choose the components. Multiple supervised machine learning classifier models were subsequently trained to utilize the ideal feature combination. The ability to make decisions was tested in three distinct ways, each with advantages. To evaluate the models' statistical significance and five-fold cross-validation, and predictive power were used. Various metrics were used to evaluate the constructed models, and their standard deviation and mean are shown below. The following is a more thorough explanation of the process, step-by-step.

## Evaluation metrics

The following measures were used to rate the supervised models' prediction abilities: True negative, true positive, false negative, and false positive rates are denoted here by TN, TP, FN, and FP, respectively.

   **ROC-AUC:** It presents an overall assessment across the entire probable classification level. "ROC" refers to the region under the receiver operating characteristic curve, or "AUC." The receiver operating characteristic (ROC) curve is generated by plotting the true

positive rate against the false positive rate, two key variables that are critical in assessing the performance of a diagnostic test or predictive model:

$$\text{True positive rate} = \frac{TP}{TP + FN}$$
$$\text{False positive rate} = \frac{FP}{FP + TN}$$

From [0, 0] through [1, 1], the entire area under the ROC curve is calculated using the AUC formula in two dimensions.

**Accuracy:** An essential parameter for classification models is accuracy. The test dataset in this study is balanced. Thus, it will indicate how well the model predicts outcomes.

$$\text{Accuracy} = \frac{TP + TN}{TP + FN + TN + FP}$$

**F1 score:** The F1-score measures a classifier's ability to detect true positives and true negatives accurately. It is mathematically represented by-

$$\text{F1-score} = 2 * \frac{\text{Presicion} * \text{Recall}}{\text{Precision} + \text{Recall}}$$

where, recall $= \frac{TP}{TP+FN}$, precision = true positive rate $= \frac{TP}{TP+FP}$

# XG-BOOST FEATURE IMPORTANCE

XG-Boost is used to determine the relative importance of the biomarkers and assess which ones impact the outcome most. XG-Boost is an effective machine learning method that determines features most useful for discriminating model outcomes (*Khan, Khan & Nazir, 2022*). By averaging a feature's relevance across all of the trees, its final importance is calculated. The frequency at which an element is utilized to split a tree is evaluated as the square of the difference in model performance brought about by the split. To ascertain the importance of features relative to the tree, we randomly selected 80% of the samples from the training set 100 times and calculated the average significance of each element (*Biswas et al., 2022*). Logistic regression has a target while setting the XGBoost regularization parameter to 5, the learning rate to 0.2, and the maximum depth to 3. The default settings for each of the other parameters are used.

## Selection of NN features

A neural network was employed to optimize mortality prediction accuracy and ascertain the optimal number of features necessary for effective predictive modeling. Artificial neural networks (ANNs) can acquire intricate interrelationships between attributes, thereby enabling the capture of complex and nonlinear patterns in data. The input layer of neurons was configured to have the same count as the number of characteristics that were being analyzed. There are two concealed layers that having Re-LU activation functions, which are present in the architecture.

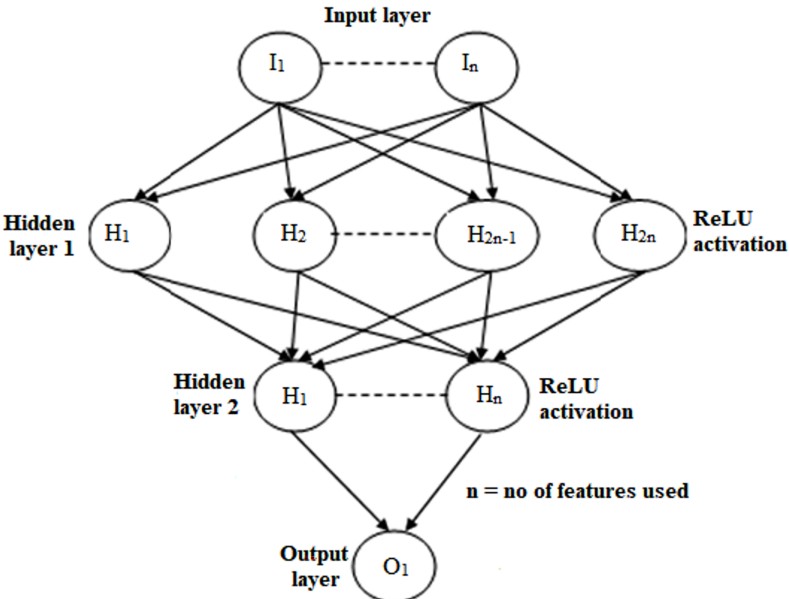

**Figure 4** **The design of the neural network used to pick features, where n is the quantity of features to be examined.**

In the neural network's topology, the second hidden layer mirrored the first hidden layer in terms of neuron count, maintaining a balanced structure in the network architecture. The neural network topology is illustrated in Fig. 4, providing a visual representation of the structural composition of the network. We utilized data from the patient sequence with the Adam optimization using a learning rate of 0.001 and a decrease in Plateau scheduling with an update in the learning rate and weights rate. Given that the NN also is not overfitting just on the training set, shallow NN should theoretically be able to perform as well as or greater than logistic regression (*Leal et al., 2022*).

**1. Neural network:** The neural network for predicting mortality has been built and used for training. The input layer is designed to have an equal number of neurons, which are utilized across multiple layers of the neural network as the best possible features selected through the feature selection process. The scheduler's training rate and patience were set to 50 to promote overfitting solely on the training data. The category cutoff was determined based on the F1-score for performance comparison metrics as assessed against the validation set. When many cutoffs produced identical F1-scores, the threshold that was nearest to 0.5 was preferred.

**2. Logistic regression:** A model with interpretability that excels with straightforward linearly separable data (*Sartorao Filho et al., 2022*; *Paul et al., 2023*). Due to the L1 penalty, limited dataset size, the threshold for halting criterion intent to 0.0001, the inverse of normalization intensity C intent to 10, and interception scaled set to 1, it was learned to use the "lib linear" solver.

**3. Random forests:** RF employs a tree-based strategy that guarantees consistency in its predictions by constructing an ensemble of decision trees that can effectively handle outliers and missing data. The data maintains accuracy while being of a tiny size (*Li et al.,*

*2021*, *2023*). It was trained using the Gini criterion, 90 trees, a minimum of one observation at a leaf node, and the maximum expected setting of nine.

**4. XG-boost:** During the training phase of XG-Boost, the following settings were used: a minimum sum of instance weight of 0 was required in each child node, a minimum learning rate of 0.2 was set, the maximum depth was 4, the full delta step was 0, the subsample size was 1, and a regularization strength of seven was applied to the L2 penalty term lambda during the training phase.

**5. Support vector machine (SVM):** SVM is a powerful machine learning algorithm that is particularly effective for small datasets and can generalize well (*Hassan, Izquierdo & Piatrik, 2021*). In a comparison of four different impacts on a variety— sigmoid, poly ("polynomial"), and RBF ("radial basis function")—the "poly" kernel outperformed the others. During the training process of the SVM model, we utilized "poly" kernels, set the gamma parameter to scaling, degrees to three, maximum iterations to 500,000, and regularization constraint C to five.

**6. Decision trees:** Decision trees utilize data to derive simple decision rules that enable them to predict the values of the dependent variable (*Saleem et al., 2022*; *Alali, Harrou & Sun, 2022*). It was learned through the following settings: splitter set to "random," maximum depth set to nine, and required sample count needed to divide a node set to two. Testing models that have been trained and validated on various folds were acquired using five-fold stratification and cross-validation (*Javed et al., 2023*). A modest dataset size necessitated using five-fold cross-validation to ensure enough variance to capture the underlying distribution. After being put to the test on the test set, the model's predictive performance was calculated by averaging the results.

**Processing test set:** The test set was subjected to min-max scaling normalization using the training set as a reference. The following two methods were used to process the rows that lacked values:

1. We deleted any rows missing data for three or more of the five attributes we chose. The next test set's missing data for each sample has been replaced using KNN with the median of those values among its related 10 closest neighbours, where the neighbors were only identified through the preferred five attributes. One approach to identifying nearest neighbours involves calculating the Euclidean distance between data points and then taking the inverse of that distance.

2. Due to the absence of values in all rows, a decision was made to remove them from the dataset, resulting in a set that exclusively contains complete cases without any imputation. It creates a test set that accurately replicates 100% of real-world testing scenarios.

Three cases are tested: The models were tried, considering three scenarios, to determine their actual prediction performance and consistency. The three cases that follow each have unique importance:

**1. Case ≤ n:** For test samples where the number of days until the outcome was n or less, any values with a frequency greater than the nth day were removed. Later, a combination of the remaining examples was made and analyzed.

**2. Case ≥ n:** Test samples were rejected when the number of days before the outcome was n or greater and the value entered for "Number of days till outcome" was smaller than n. Next, the remaining samples were merged, and the results were evaluated.

**3. Case = n:** In this case, n days exactly before the result, $n^{th}$ day's value for "number of days till outcome" was used to choose test samples.

## RESULTS

For the recognition of important biomarkers for evaluating patients' medical conditions, some factors are required, such as the fact that many lab tests are needed, especially given the severity and sudden increase of SARS-CoV-2 infections and related mortality. Feature selection, which aids in identifying the much more sensitive and significant biomarkers that assist risk evaluation, reducing the number of lab tests ordered is crucial as it can expedite decision-making and improve overall efficiency.

**NN feature selection:** forward sampling that used a neural network to plot the AUC values of each set of attributes. This task aimed to lower the number of features utilized in the model while raising the AUC score. The average AUC score across five facts was found to be 0.95. The addition of the sixth characteristic did not result in a significant increase in the area under the curve (AUC) metric.

Figure 5 presents the observed accuracy in relation to the number of selected features utilized for the modelling purpose. Omission of the "age" feature resulted in a decrease in the performance of the model, although the model's performance did not change when the "gender" addition was added. Accordingly, the traits chosen for this investigation are hs-CRP, age, LDH, neutrophils (%), and lymphocytes (%). The statistical significance of these desired traits is also examined using the ANOVA f-test (*p*-value 0.001). The model's capability to predict six distinct techniques—neural net, XG-Boost, logistic regression, SVM, decision trees, and random forests—has been used to develop the classification models. Following filtered five-fold cross-validation, each method was first assessed using samples from all days of the test set. Figure 6 displays the F1-score, AUC, and the accuracy of each developed model created utilizing each of the six techniques. When effectively discriminating between the two groups, we want to select the most precise estimation possible. The neural network was the best model since it outperformed the others in terms of accuracy (96.54%); by examining three separate scenarios, the model's applicability and resilience in various contexts were further tested (Cases 1, 2, and 3).

When n or fewer days remain until the day of the outcome, the first scenario examines the model's performance when the test set only includes data (Fig. 7). In testing, many n values were taken into consideration. According to Fig. 8, the precision remains persistently exceptional for n up to 3 weeks, provided that there are n or fewer days until the occurrence of the outcome. The accuracy rates demonstrated the model's potent predictive ability, which fell within a small range of 97.19–99.1% once again for values of n from 0 onwards 17 days.

The analysis is done for only 17 days because relatively few sample sizes after day 17 can impact this precision later. Fig. 9 shows high and consistent F1-scores (worst = 0.97 as well as best = 0.99) and AUC scores (worst = 0.99 as well as best = 1) over a range of days.

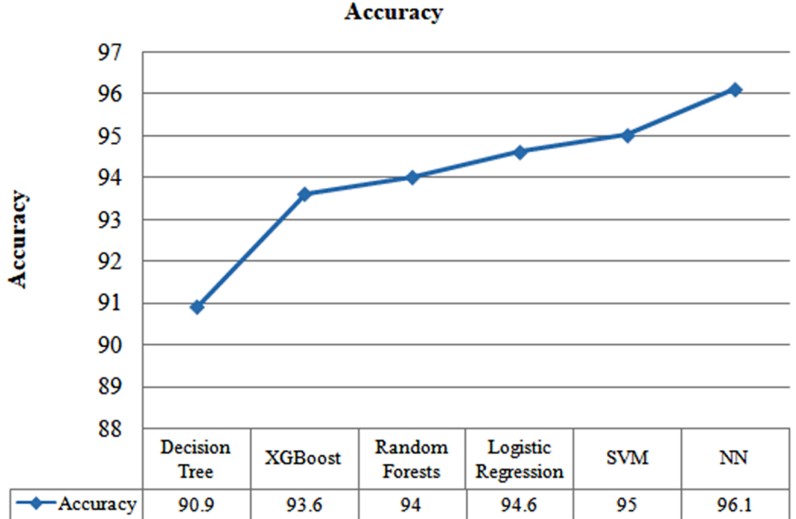

**Figure 5** Performance evaluation of several machine learning algorithms using accuracy.

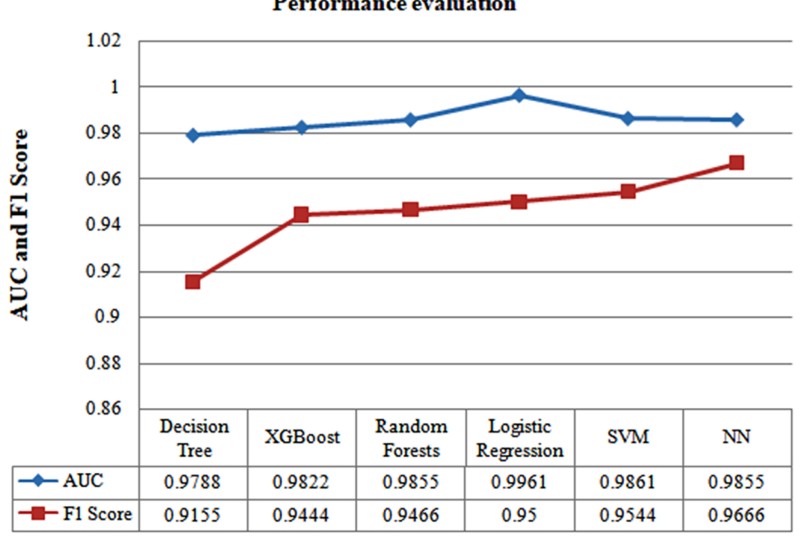

**Figure 6** Performance evaluation of several machine learning algorithms using F1-score and AUC.

Demonstrates that the model consistently operates at least 97% accurately, regardless of how many days remain before the result. The disadvantage of the base case is that it contains more samples closer to the outcome, which increases the likelihood, that they will dominate the results.

Case 2, which evaluates the model's performance for n or perhaps more days, was considered to address this issue. Due to the unavailability of predictions of outcomes after a certain date, to ensure an accurate evaluation of the model's performance during an illness, it is recommended to conduct retrospective validation by evaluating the model's predictions on prior days leading up to the outcome day.

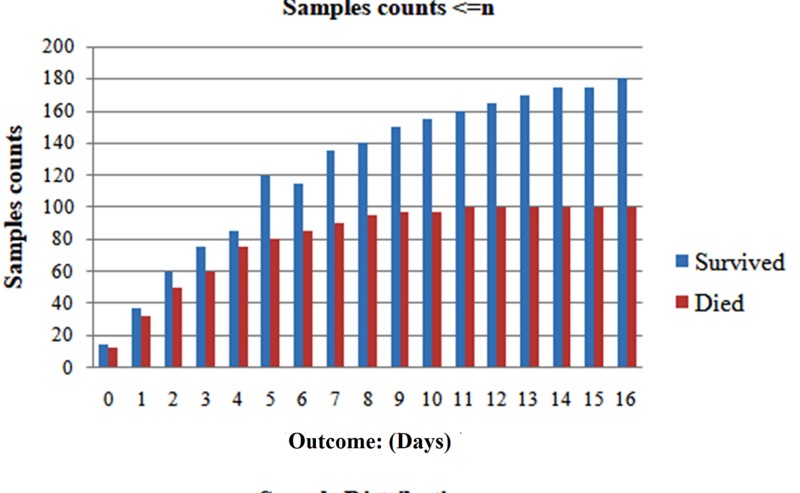

**Figure 7** Analyzing case 1's test data, a neural network's performance: shorter than or similar to n days till the result, as the cumulative data points including all samples inside this imputed test set.

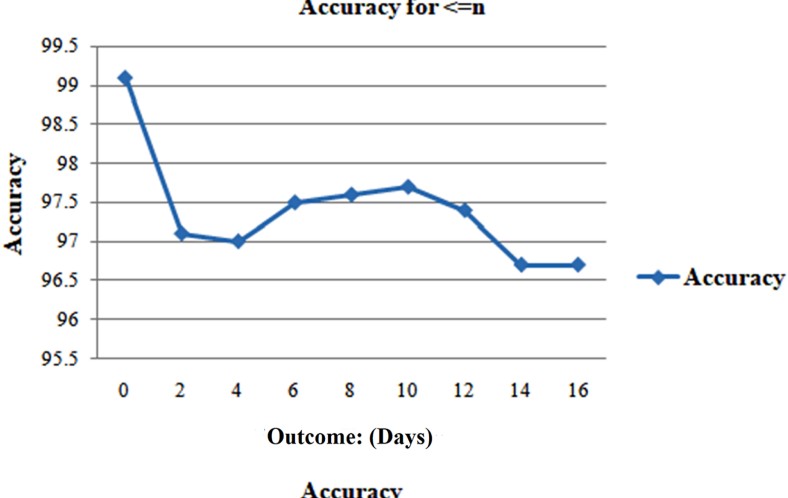

**Figure 8** Analyzing case 1's test data, a neural network's performance: shorter than or similar to n days till the result, as evaluation of the model's accuracy for various days' results.

The collection of samples closest to the outcome day should influence all cumulative computations due to the reduction in the number of samples as the period before the effect increases, as depicted in Fig. 10. Consequently, this provides a more accurate summary. It makes it tougher to forecast the results as we approach closer to the outcome day, which is supported by Fig. 11. The model exhibits that at the outset of the outcome day, the model attains an exceptional accuracy of 96.6%, sustained consistently throughout the evaluation period, maintaining a high level of precision. When the model makes predictions 15 days or more in advance, the accuracy drops to 87.8%. The AUC and F1 scores displayed a

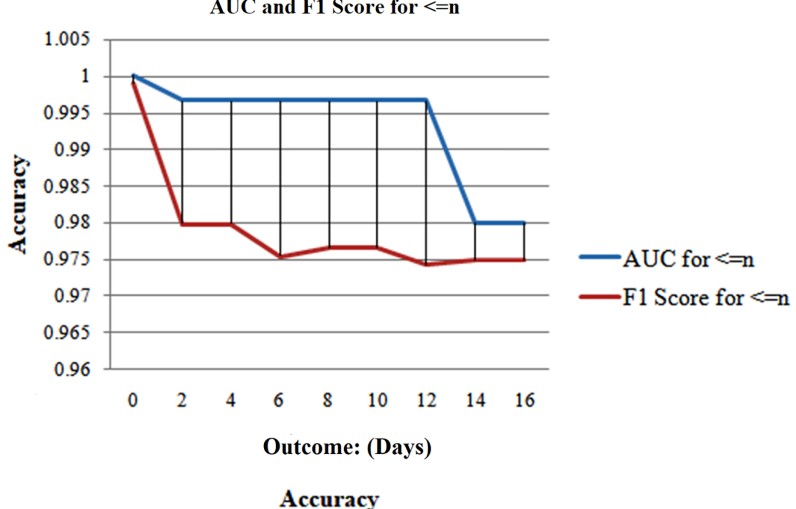

**Figure 9** Analyzing case 1's test data, a neural network's performance: shorter than or similar to n days till the result, as AUC and F1-score of such model were examined for various days to result.

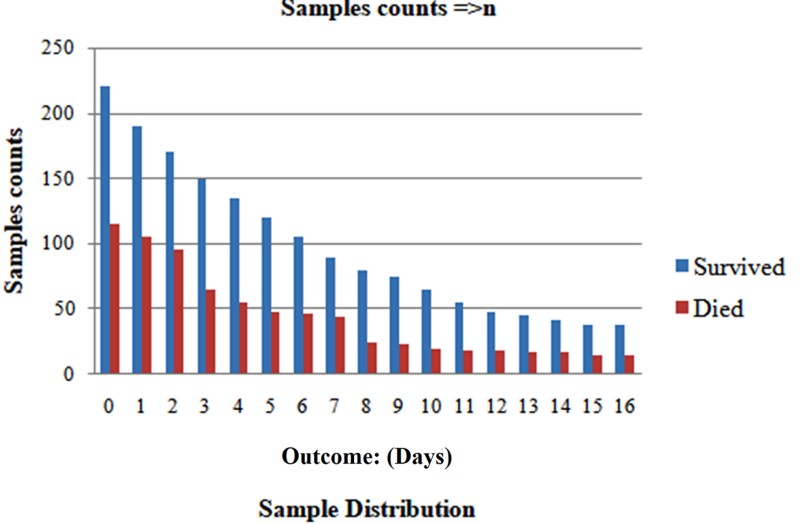

**Figure 10** Analyzing case 1's test data, a neural network's performance: greater than or similar to n days till the result, as the cumulative data points including all samples inside this imputed test set, distributed per class n^th Days.

comparable character, as seen in Fig. 12. On the day of the outcome, the AUC starts at 0.99 and drops to 0.97 on day 15. F1-scores begin at 0.98 and drop to 0.85 on day 13 to their lowest position.

But when n is less than 1 week, the model performs well and predicts with a good accuracy of 95.8%. The model's performance is then evaluated in Case 3 for even more thorough testing, which ensues precisely n days before the outcome day (Fig. 13).

The result in this case cannot be influenced by any sample from a different day. Figure 14 demonstrates that the model suggests the effect with a precision of 98.8% when it

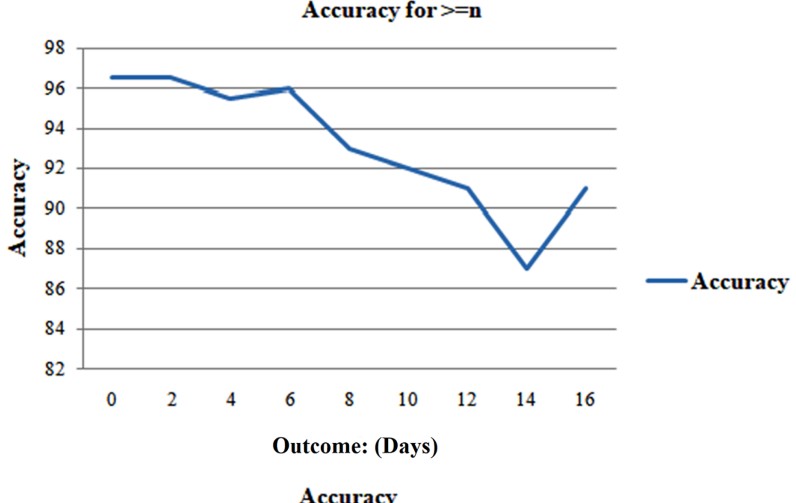

**Figure 11** Analyzing case 1's test data, a neural network's performance: greater than or similar to n days till the result, as evaluation of the model's accuracy for various days' results.

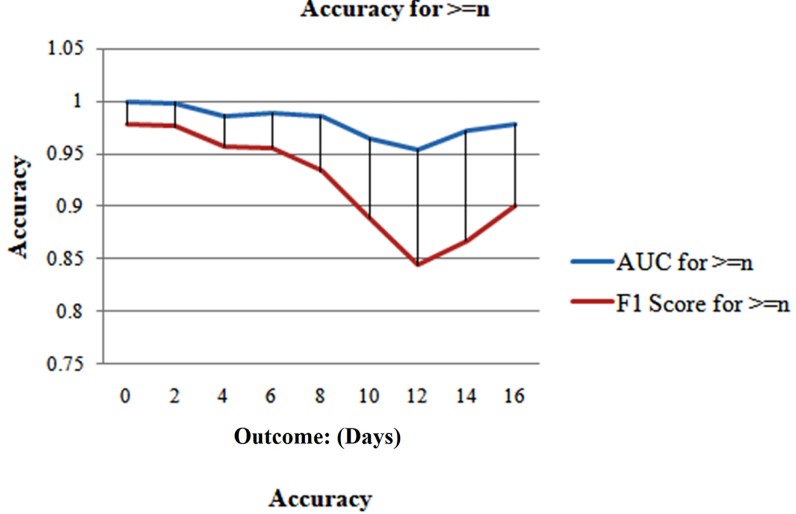

**Figure 12** Analyzing case 1's test data.

comes to the consequence and gains its minimum value of 92.86% on day 5. Although the accuracy varies over the weekdays, Fig. 14 illustrates that the accuracy swings during the days.

The F1 scores and AUC follow a similar pattern, with the former beginning at 1 and the latter at 0.99 on the first day, respectively, reaching their lows of 0.95 and 0.95 on day five (Fig. 15). Given the smaller sample size, it's probable that 100% accuracy for days 7–12 was also caused by this. But it's crucial to remember that both groups were still there, and the model consistently identified the F1-scores and the AUC at 1 for days 7 to 12 with accuracy. The neural network demonstrated excellent performance on the test set, with an efficiency of 85%, an AUC of 98%, and a total F1-score of 95%. The next best method,

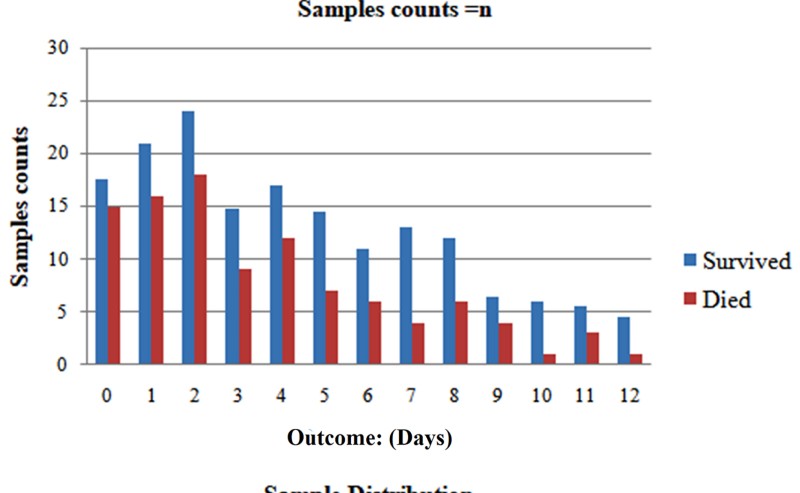

**Figure 13  Analyzing case 3's test data, a neural network's performance: equal to n days till the result, as the cumulative data points including all samples inside this imputed test set, distributed per class (nth Days).**

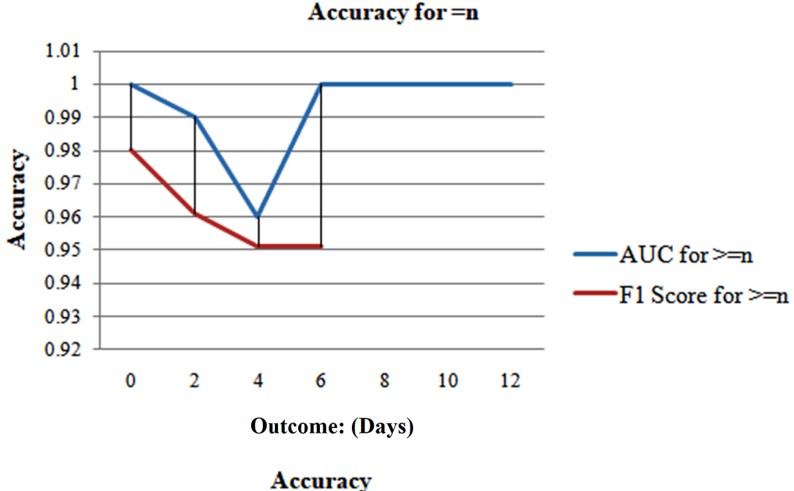

**Figure 14  Analyzing case 3's test data, a neural network's performance: equal to n days till the result, as the cumulative data points including all samples inside this imputed test set, distributed per class (nth Days).**

logistic regression, achieves results comparable to those of neural networks. We once again more employed these three methods (cases 1, 2, and 3) as described previously to assess the predicted performances and undertake robust testing.

## DISCUSSION

Using neural networks and XG-Boost feature significance, this investigation aimed to balance selecting only a limited number of characteristics and achieving a high AUC score for constructing the models. By adopting this method, a robust and effective combination of five features has been identified for predicting mortality, thus reinforcing the model's

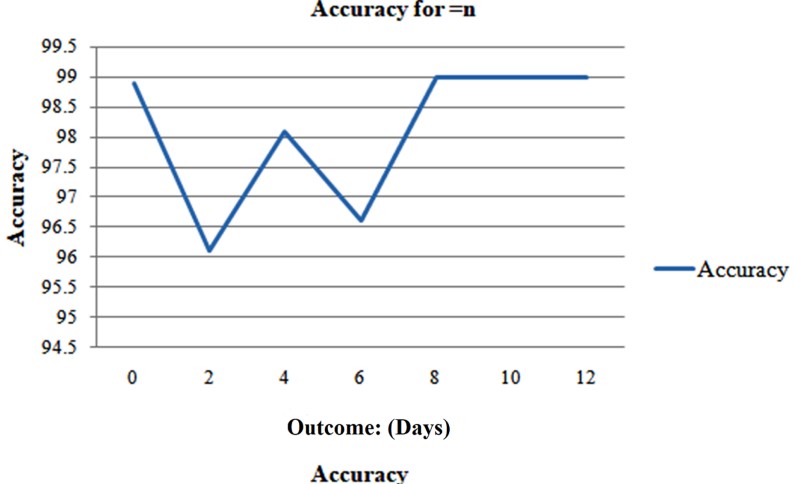

**Figure 15 Analyzing case 3's test data, a neural network's performance.**

predictive capabilities. The five desired characteristics are LDH, hs-CRP, neutrophils (%), age, and lymphocytes (%). These traits have all been discovered to be predictors of death from COVID-19 disease (*Myoung, 2022*; *Arslan & Arslan, 2021*; *Biswas et al., 2022*; *Li et al., 2021*; *Saleem et al., 2022*). All of the models utilized here reflect that age has been shown to significantly contribute to the progression of COVID-19 disease (*Mousavizadeh & Ghasemi, 2021*; *Li et al., 2022*; *Ray et al., 2021*; *Arslan & Arslan, 2021*; *Li et al., 2021*; *Saleem et al., 2022*).

Patients under 60 had a higher failure rate. They needed longer hospital admissions than patients over 60 (*Mousavizadeh & Ghasemi, 2021*), which may indicate that the elderly did not respond as well to treatment as the younger generation did. Individuals over 80 years old are at a significantly higher risk of developing a serious or life-threatening illness following a COVID-19 infection, with a risk percentage of 41.4%. In contrast, those under 20 years of age have a much lower risk of only 4.1% (*Li et al., 2022*). According to studies, patients with COPD (chronic obstructive pulmonary disease) were identified as having higher hs-CRP levels than the general population (*Mohsan et al., 2022*; *Madhav & Tyagi, 2022*). The hs-CRP levels correlate with worse pulmonary function (*Dubey, Verma & Kumar, 2024*; *Hu et al., 2022*). In the investigation of the classification of COVID-19 patients, it has also been discovered that the hs-CRP is a crucial component (*Ratajczak & Kucia, 2021*). Elevated levels of hs-CRP point to the emergence of severe microbiological re-infection that is anticipated as a frequent result in serious infected patients and may increase death. Finding patients in danger of super infections or other issues may help treat them more effectively before large rises in LDH and hs-CRP levels are apparent (*Apostolopoulos & Mpesiana, 2020*).

A kind of WBC (white blood cell) known as a neutrophil is the initial line of defence in inflammation (*Marques, Agarwal & de la Torre Díez, 2020*). A body may be contaminated if neutrophil counts are elevated. When compared to those who survived, individuals who died had noticeably higher neutrophil counts (%). T cells, NK cells, and B cells—well-

known—are components of lymphocytes. Again, we identified that the percentage of lymphocytes in the blood of those who died was much lower than that of those who lived. The NLR is a predictive marker for severe infection among infectious disease sufferers like COVID-19 (neutrophils-to-lymphocyte ratio) (*Wang et al., 2021*).

The outcomes of the study show significant improvements from previous investigations in a number of important domains. First off, compared to other methods of imaging such as ultrasounds, CT scans, and X-rays, routine blood tests are more widely available, making their use for infectious diseases like COVID-19 fatality predictions an important development. The accessibility issue that exists in numerous medical facilities is addressed by this method. The efficacy of this attribute choice is demonstrated by the ML (machine learning) model's remarkable 96% accuracy upon the inclusion of five different traits (lymphocytes, neutrophils, LDH, age, and hs-CRP). This shows the value of those blood-based biomarkers in morbidity forecasting and outperforms the predictive ability of earlier research.

The study thoroughly assesses a range of machine learning models, such as logistic regression, neural networks, random forests, XG-Boost, decision trees, and SVM model, in the context of model comparison. This study stands out for its unambiguous conclusion that neural networks and XG-Boost are the top-performing techniques, offering a helpful suggestion for medical professionals and investigators looking for trustworthy algorithms for morbidity predictions. The suggested model outperforms previous methods with regard to dependability and precision, as evidenced by the remarkable enhancement in accuracy in predicting it achieves, particularly when using neural networks and XG-Boost.

The revolutionary finding of the research is its ability to predict fatalities up to 16 days ahead of the final result, outperforming the time-dependent predicting abilities of prior approaches. In the context of infectious disease, when prompt and efficient decisions are critical, this immediate detection provides for best practices and prompt treatment for patient care. The suggested model's viability and validity have been reinforced by the thorough testing conducted on a variety of instances, enhancing its relevance to actual-life circumstances.

In conclusion, the research's methodology is excellent because it makes use of data from ordinary blood tests, optimizes selecting features, thoroughly assesses different machine learning (ML) models, achieves high accuracy in prediction, and generates predictions early on. Together, these improvements establish the proposed system as a cutting-edge approach in the domain, outperforming previous research and providing an effective foundation for mortality associated with infectious diseases like COVID-19 prediction.

## CONCLUSION

A potent union of five independent variables was discovered due to the research detailed in this article: neutrophils (%), age, lymphocytes (%), LDH, and hs-CRP, which can be used to accurately predict a patient's death from an infectious disease like COVID-19. Different learning models, including mortality prediction, have been created for the best comparison. The neural network provides a framework for fatalities: 96% for any day during the disease and 90% for more than 16 days in the future. Model interpretability that

might be useful in a clinical environment can be provided by XG-Boost feature significance. The recommended model's robustness has been tested in three separate instances. The proposed model is very confident and grateful for the acquired performance measurements. Several relevant predictors discovered that additional information from multiple places is needed to confirm their applicability and improve the model.

The accuracy and effectiveness of any data-driven methodology, especially those based on machine learning algorithms, are significantly influenced by the quality of the input data. The presence of inaccurate or incomplete data can adversely impact the performance of the model, leading to erroneous predictions. Additionally, data quality issues such as the existence of missing values, outliers, or measurement errors can further undermine the model's accuracy and decrease its predictive power. Hence, ensuring the quality of input data is a critical consideration for any machine learning-based approach in the healthcare domain, given the importance of accurate predictions for decision-making in clinical settings.

## ACKNOWLEDGEMENTS

The authors express their gratitude to the Jaypee University of Engineering Technology, located in Guna, Madhya Pradesh, India, for providing them with infrastructural support. Additionally, the authors extend their thanks to all members of the DPMC for their valuable guidance and support. Their contributions were instrumental in the success of the research, and the authors acknowledge their efforts with deep appreciation. The authors recognize the importance of collaborative efforts and support in achieving their research goals and hope that such initiatives will continue to flourish in the future. In their research, the authors had access to Genome data that was made available as open access repositories. The authors express their appreciation for the valuable contributions of all individuals who have provided these services. The availability of such data allowed the authors to conduct their research and contribute to the advancement of knowledge in their field. The authors recognize the importance of open access repositories in promoting scientific collaboration and improving access to critical research data.

### Funding
The authors received no funding for this work.

### Competing Interests
The authors declare that they have no competing interests.

### Author Contributions
- Shivendra Dubey conceived and designed the experiments, performed the experiments, analyzed the data, performed the computation work, prepared figures and/or tables, authored or reviewed drafts of the article, and approved the final draft.

- Dinesh Kumar Verma conceived and designed the experiments, performed the experiments, analyzed the data, prepared figures and/or tables, and approved the final draft.
- Mahesh Kumar conceived and designed the experiments, performed the experiments, performed the computation work, authored or reviewed drafts of the article, and approved the final draft.

## Data Availability

The dataset is available at Zenodo: DUBEY, S. (2024). Data set for genome sequence and mutation rate analysis [Data set]. Zenodo. https://doi.org/10.5281/zenodo.10682056.

The code is available at Zenodo: DUBEY, S. (2024). Models for the Genome Sequence Analysis and Mutation Rate Analysis. Zenodo. https://doi.org/10.5281/zenodo.10682261.

## Supplemental Information

Supplemental information for this article can be found online at http://dx.doi.org/10.7717/peerj-cs.2062#supplemental-information.

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
