# Peer review of "Real-time infectious disease endurance indicator system for scientific decisions using machine learning and rapid data processing"

_PeerJ Computer Science, doi:10.7717/peerj-cs.2062_

## Round 0.1 · original submission · Major Revisions

Please consider the reviewers comments.

**Language Note:** The review process has identified that the English language must be improved. PeerJ can provide language editing services - please contact us at [email protected] for pricing (be sure to provide your manuscript number and title). Alternatively, you should make your own arrangements to improve the language quality and provide details in your response letter. – PeerJ Staff

Reviewer 1 ·

Basic reporting

I did not identify any new features, or open or resolved problems, nor are the scientific contributions clear. Computationally, which metrics, attributes, and criteria were improved about what already exists?

Experimental design

The criteria used, the parameters, and the results obtained are not clear.

Validity of the findings

The criteria used, the parameters, and the results obtained are not clear.

Additional comments

I strongly suggest that authors review the related works section to better clarify the criteria used. The implementation carried out, the tests, and the results obtained are also not clear how they arrived at the values, in short, the topic is interesting but there are several gaps.

Reviewer 2 ·

Basic reporting

The authors propose a novel scheme for leveraging machine learning (ML) methodologies to predict the likelihood of mortality in infectious diseases like COVID-19 based on blood test data. They identify five highly impactful features – age, LDH, lymphocytes, neutrophils, and hs-CRP – which, when combined, achieve an impressive 96% accuracy in predicting mortality. By integrating XGBoost feature importance with neural network classification, the optimal method further elevates the predictive power by achieving exceptional accuracy and 90% precision in mortality prediction for infectious diseases, up to 16 days before the event. The authors demonstrate the model's robust performance and practical applicability through testing with three instances based on varying timeframes leading to the outcome. Although promising, the approach necessitates further enhancements before it can be fully embraced.

Choice of five features: The paper does not explicitly explain the choice of five specific features. Please justify their selection or suggest relevant literature that highlights the significance of these features.
Missing key contributions: The introduction lacks clarity on the study's specific contributions. Please mention them explicitly early on.
Literature review: A table comparing the proposed approach with existing literature would be beneficial for clarity and impact.
Number of biomarkers: 74 biomarkers might be insufficient for a comprehensive evaluation. Authors should justify their choice or explore adding more relevant markers.
Data set characteristics: The description of "Name, age, gender, address, contact information, etc." is imprecise. Authors should provide a detailed list of all data set characteristics, including classes, to avoid ambiguity.
Data set table: A table summarizing key insights about the data set, including classes, would be valuable.
Figure quality and results discussion: Poor figure quality and insufficient result discussion weaken the paper's persuasiveness. The authors should improve figure quality and explicitly discuss each result, highlighting how their scheme surpasses existing studies in specific areas.
By addressing these points and implementing the suggested improvements, the authors can significantly strengthen their research and increase its potential for acceptance.

Experimental design

Included in Basic Reporting

Validity of the findings

Included in Basic Reporting

Additional comments

None

·

Basic reporting

The work under review concerns the implementation of machine learning in rapid data processing of the COVID tests. The work is well written, readable and might be interesting for many different professionals and non-professionals.

Experimental design

The work is well-planned and the results are carefully presented.

Validity of the findings

The results are reasonable and prove the idea of the authors.

Additional comments

However:
- References are not cited in increasing order.
- The work needs additional editing. For example, there are wrongly used capital characters, missing punctuation, not well edited captions, part of the sentences are long and have might not be understandable or have wrong word order etc.
- The quality of the figures is low.
- The caption of Figure 5 is missing.
- The positions of the figures is not shown in the work.

---

## Round 0.2 · Minor Revisions

The figures are not readable - please provide legible versions.

·

Basic reporting

The work under review estimates the risk during COVID. Generally speaking, the research is interesting and correlates the outcomes and the treatment.

Experimental design

The experiments are well-planned and conducted.

Validity of the findings

The findings might be interesting for the public.

Additional comments

I think the quality of the figures is low.

---

## Round 0.3 · accepted · Accept

Based on the previous reviews, the manuscript was accurately revised. Also, the statistical analysis performed is adequate to the study.